# ECM-mediated positional cues are able to induce pattern, but not new positional information, during axolotl limb regeneration

**Warren A. Vieira, Shira Goren, Catherine D. McCusker***

Department of Biology, University of Massachusetts, Boston, MA, United States of America

* Catherine.McCusker@umb.edu

## Abstract

The Mexican Axolotl is able to regenerate missing limb structures in any position along the limb axis throughout its life and serves as an excellent model to understand the basic mechanisms of endogenous regeneration. How the new pattern of the regenerating axolotl limb is established has not been completely resolved. An accumulating body of evidence indicates that pattern formation occurs in a hierarchical fashion, which consists of two different types of positional communications. The first type (Type 1) of communication occurs between connective tissue cells, which retain memory of their original pattern information and use this memory to generate the pattern of the regenerate. The second type (Type 2) of communication occurs from connective tissue cells to other cell types in the regenerate, which don't retain positional memory themselves and arrange themselves according to these positional cues. Previous studies suggest that molecules within the extracellular matrix (ECM) participate in pattern formation in developing and regenerating limbs. However, it is unclear whether these molecules play a role in Type 1 or Type 2 positional communications. Utilizing the Accessory Limb Model, a regenerative assay, and transcriptomic analyses in regenerates that have been reprogrammed by treatment with Retinoic Acid, our data indicates that the ECM likely facilities Type-2 positional communications during limb regeneration.

## 1. Introduction

Human limb loss can result directly from trauma or secondary to diseases, such as cancer and diabetes, and is associated with many physical, psychological, and financial burdens on the patient [1–3]. One of the major goals of regenerative medicine is to engineer a means to repair missing limb tissues endogenously within the human body. A variety of model organisms, which unlike humans exhibit robust regenerative ability, are utilized to assess the requirements and mechanisms of innate regeneration in the hopes of achieving this goal.

Limb regeneration has been extensively studied in the Mexican axolotl (*Ambystoma mexicanum*), a urodele amphibian. This process involves the formation of a transient regenerative organ known as the blastema, constituted by dedifferentiated limb progenitor cells and the appropriate molecular signals. Together these cells and signals control the growth, patterning,

**Funding:** This work was funded by the Eunice Kennedy Shriver National Institute of Child Health and Human Development of the Nation Institutes of Health [grant number 1R15HD092180-01A1], awarded to CDM. The funders had no role in study design, data collection and analysis, decision to publish, or preparation of the manuscript. Funder website: https://www.nichd.nih.gov/.

**Competing interests:** The authors have declared that no competing interests exist.

and differentiation of the blastema into the missing limb structures. This overall regenerative process has been well characterized as having three basic biological requirements [4]. The first two requirements, the establishment of a wound epithelium and the innervation thereof, are essential for the generation of a regeneration permission environment [4]. This environment is sufficient for initial blastema formation, where stump derived cells revert to a more progenitor-like cell state in order to provide the building blocks for regenerative output. However, this environment is insufficient, on its own, to generate the blueprint, or pattern, of the missing limb structures [4]. The third and final requirement for limb regeneration is a positional disparity that is established between cells that arise from the different axes of the limb. These cells are derived from connective tissue origins in the stump [4–6], migrate into the regenerative permissive environment [4, 7, 8], and play an essential role in driving the formation of the missing limb pattern.

The sufficiency of the above listed requirements, a wound epithelium, a nerve, and a positional disparity, for limb regeneration is exhibited through the regenerative assay known as the Accessory Limb Model (ALM) [4]. In this assay a complete ectopic limb is formed from a wound site generated on the side of the limb by surgically 1) removing a square of full thickness skin, 2) deviating a limb nerve bundle into this wound site, and 3) grafting cells from the opposite side of the limb, relative to the wound site [4, 9, 10], or treating with specific signaling molecules [11, 12]. One of the strengths of the ALM is that it allows us to perform "gain of function" assays to test the molecular mechanisms that drive each step of the regenerative process. For example, this assay was used to show that the extracellular matrix (ECM) carries some positional information [13], to identify combinations of growth factors that suffice for nerve signaling [14, 15], and to generate complete limbs from lateral wounds that were treated with signaling molecules at different time points [7].

However, the nature of the endogenous signals that connective tissue cells use to generate new pattern remain elusive. There are multiple models that explain how new limb pattern emerges—some are based on direct cell-cell interactions between pattern forming cells [16, 17]; others are based on the generation of morphogen gradients by these cells [18–20], and still others are based both on cell-cell interactions and the generation of signaling centers by these cells [21]. In our recently proposed model, known as the Patterning Hierarchy Model, we propose that patterning of the regenerate is dependent on two types of positional communications. The first type, Type 1, occurs between the connective tissue cells that are derived from the different limb axes in the stump, which establishes the pattern of the regenerate 4,5,12,33–36]. Once the basic limb pattern is established, the second type of communication, Type 2, occurs between the connective tissue derived cells and the other limb progenitor cells, to arrange them into their final conformation [12, 22–24]. Although a number of signaling molecules and proteins have been implicated in pattern formation during regeneration [11, 25–29], determining the type of positional communication that these proteins provide will be an essential step to understanding their role in patterning.

Growing evidence suggests that the ECM participates in the communication of positional cues during pattern formation. Utilizing the ALM assay, it was shown that ectopic limb tissue could be induced by implanting decellularized ECM from the posterior side of the limb into an anterior-located innervated limb wound site [13]. Although the overall pattern of these regenerates lacked complexity, often forming single nodules of cartilage that were tapered distally, grafts of anterior skin into the same wound locations fail to generate similar structures [13]. Thus, the ECM tissue itself must contain molecules that provide positional cues in the regenerating environment.

Although there are many types of molecules present throughout the ECM, Heparin Sulfate Proteoglycans (HSPGs), are strong candidates for playing a role in positional communications

during regeneration. HSPGs are located on the cell surface or within the ECM, and are constituted by long, linear glycosaminoglycan heparan sulfate (HS) chains covalently linked to a core protein and can act as receptors, co-receptors, recruiters, and ligand sinks [30, 31]. Multiple signaling pathways that play key roles in limb patterning, including fibroblastic growth factor (FGF), WNT, and Hedgehog (as reviewed by [30, 32]), are regulated by HSPGs. Additionally, heparan sulfate sulfotransferases (HSST) genes, responsible for modifying HS chains, are hypothesized to contribute to limb patterning during embryogenesis [33]. The implantation of an artificial ECM constructed of type 1 collagen and HS alone is capable of generating small ectopic structures in the context of the ALM [13]. Additionally, the complexity of the resulting limb pattern in anterior ALMs implanted with posterior-derived connective tissue is reduced when the tissue is treated with heparinase-III, thereby removing HS side chains, prior to grafting [5]. These findings suggest that HSPGs contribute to, but are not solely responsible for, pattern formation during regeneration.

Given the evidence that HSPGs play a role in providing positional cues, we sought to determine what type of cue they constitute–a Type 1 cue that generates cells with new positional information or a Type 2 cue that communicates this positional information to the other tissues in the regenerating limb. Our data, obtained from the ALM and expression analyses, indicates that HSPGs fall into the latter category of positional communication.

## 2. Results

### 2.1. ECM-induced structures failed to regenerate in response to amputation

According to the Patterning Hierarchy Model of regeneration, Type 1 positional communications occur between connective tissue derived cells (i.e. pattern forming cells) in the blastema in order to generate the missing positional information to complete the pattern of the regenerate. Type 2 communications then dictate how the other cells (i.e. pattern following cells) should arrange themselves in this context. Thus, one way to decipher between Type 1 and Type 2 communications is to test whether they have generated new positional information in the regenerating tissue or not. We assessed whether new positional information had been generated in ECM-induced ectopic limb structures using two assays. In the first, we assessed the regenerative ability of the limb structures that were induced by a posterior ECM graft into an anterior ALM. If these structures regenerate, then this implicates that new, stabile, positional identities were induced by the ECM in the ALM. In the second assay we grafted the tissue that was excised from these ECM-induced structures into both anterior and posterior located ALMs. If these grafts result in the formation of ectopic limb structures in both ALM locations, then this would additionally indicate that new, stabile, positional information is generated by the ECM grafts. However, if ectopic limbs were only generated in posterior located ALMs, then this indicates that no new positional identities were generated in the ECM-induced structures.

We utilized the modified ALM-protocol, described by Phan et al. [13], to generate ECM-induced structures. Briefly, posterior ECM was collected from the axolotl forelimb and decellularized by treatment with urea. This acellular tissue was then grafted into anterior, innervated wound sites on the forelimb. These wounds were allowed to heal and were monitored for regenerative output. Of the 50 anterior ALM sites implanted with posterior derived ECM, live-assessment showed 37 with small, rounded or digit like outgrowths by week 9 (Fig 1, preamputation). 34 of these structures were large enough to be amputated and subsequently monitored weekly for a regenerative response (Fig 1A).

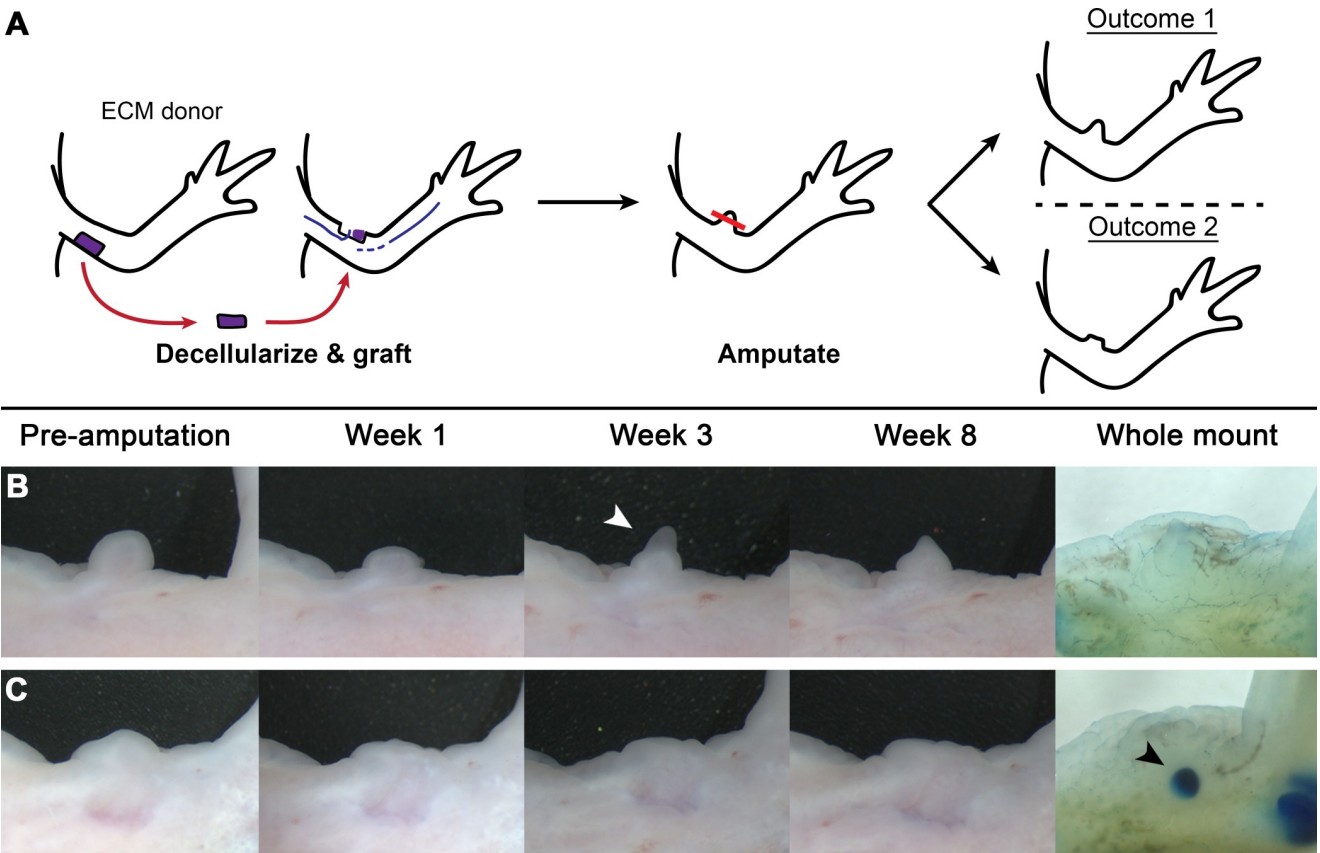

**Fig 1. ECM induced ectopic structures failed to regenerate in response to amputation.** A) Experimental design. Decellularized posterior ECM (purple) was implanted into anterior, innervated wound sites. The resultant ectopic structures were allowed to pattern and differentiate, and then amputated and monitored for a regenerative response. If decellularized ECM grafts were sufficient to elicit the generation of new (non-anterior) positional identities in the ectopic structures, then the structures would have a diversity of positional information and would regenerate in response to amputation (outcome 1). If the ECM grafts were insufficient to elicit the generation of new positional identities, then the structures would fail to regenerate as they would be constituted by only anterior positional information (outcome 2). B-C) Representative images of ECM-induced ectopic structures before and after amputation. Amputated structures were monitored for 8 weeks for a regenerative response, and two different phenotypes were documented. Panel B: In response to amputation a blastema was generated (white arrow); however, this blastema regressed by week 8. Victoria blue stained whole mounts showed the absence of any new skeletal elements in the ectopic structures at the site of blastema formation. 3 of the 34 amputated structures exhibited this phenotype. Panel C: In response to amputation no blastema was generated in 31 out of 34 of the limbs. At week 8, all 34 limbs were harvested for Victoria blue whole mount staining and the resultant whole mounts showed minimal, if any, skeletal patterning in the stump of the ectopic structures. The cartilage nodule (black arrow) present in the whole mount stained limb of panel C was situated in the stump of the ectopic structure. This pattern was generated in response to the ECM-grafted into the innervated anterior wound site.

In response to amputation of the ECM-induced structures, blastema formation was documented in only 8.8% of the cases; and these blastemas ultimately regressed (Fig 1B). As a disparity in positional information is one of the necessary biological requirements at the site of injury for regeneration [4], the absence of regenerative output in response to amputation in this experimental context suggests that ECM-induced ectopic structures lack sufficient positional diversity to stimulate a disparity. The remainder of the amputated structures generated a wound epithelium and healed over without blastema formation, as seen with simple lateral wounding (Fig 1C) [4]. These observations indicate that the ECM grafts did not induce the formation of new, stable, positional information in the structures elicited from anterior inverted wound sites.

## 2.2. ECM-induced structures are composed of tissue with an anterior, but not posterior, positional identity

To further validate the above findings, we next tested the inductive abilities of the ECM-induced structures in ALMs located on either the anterior or posterior sides of the limb. As explained above, if the donor tissue induces the formation of ectopic limbs in both locations then that indicates that both anterior and posterior positional information has been stabilized in the grafted tissue. Alternately, the induction of limbs in only posterior ALMs would indicate that the donor tissue was composed entirely of anterior positional information, and that the ECM grafts failed to generate new, stabile, (non-anterior) positional information.

We harvested and subsequently grafted full thickness skin from the distal end of ECM-induced ectopic structures from the donor animals into innervated anterior and posterior ALM sites on the host limbs (Fig 2A). Of the ALM surgeries that were performed, graft retention and blastema formation were observed in 29 anterior ALMs and 12 posterior ALMs. Posterior ALMs typically have a lower technical success rate because the wound position results in rubbing of the surgery site on the flank of the animal, which can dislodge the grafted tissue. We found that ECM-induced tissue elicited complex pattern, including complete limbs, when grafted into posterior located ALMs (75%) (Fig 2B, Table 1). However, when this tissue was grafted into anterior ALMs, only 4 (13.8%) gave rise to a structure (Fig 2C, Table 1), and the pattern of these structures were simple, single nodules of cartilage. Together these findings indicate that ECM-induced structures are composed mostly of anterior identities and have little to no posterior positional information. These observations indicate that grafts of ECM into the original anterior ALMs did not result in the formation of new, non-anterior positional information in the ectopic structures. Thus, it is unlikely that the ECM, and any of the molecules therein, constitute a Type 1 positional communication.

## 2.3. Expression of HSST genes in anterior and posterior ALMs is unaffected by RA treatment

Previous studies indicate that HSPGs play a role in patterning developing and regenerating limb structures [5, 13, 30, 33, 34]. HSPGs can be located both on the cell surface and throughout the ECM. It has been hypothesized that differences in the type, number, or sulfonation patterns of HSPGs between different limb axes directly communicates positional information to pattern the regenerating limb structures [13, 35]. HSPG sulfonation patterns are established and maintained by heparan sulfate sulfotransferases (HSST) and semi quantitative methods show an axis specific differential expression pattern for a variety of axolotl HSST within blastema tissue [13]. It was previously shown that the HSSTs *HS6ST1* and *NDST2* had elevated expression in the anterior side of the blastema, relative to posterior; while *HS3ST1* had the inverse expression pattern [13]. We hypothesized that if HSST expression directly contributes to positional identity, then limb blastema cells that are reprogramed to a different positional identity should have corresponding changes in HSST expression patterns.

To test this idea we utilized Retinoic acid (RA) treatment, which is sufficient to reprogram undifferentiated blastema tissue to a posterior-ventral-proximal identity [7, 11, 25–29]. At the molecular level, RA treatment significantly alters the expression of anterior and posterior markers. For example, *Shh* (posterior marker) expression increased and *Alx4* (anterior marker) expression decreased significantly in anterior ALMs when treated with RA [7]. Therefore, if HSSTs play a direct role in communicating positional information during regeneration, then their expression should be consistent with posterior blastema tissue (low *HS6ST1* and *NDST2* and high *HS3ST1)* following RA treatment.

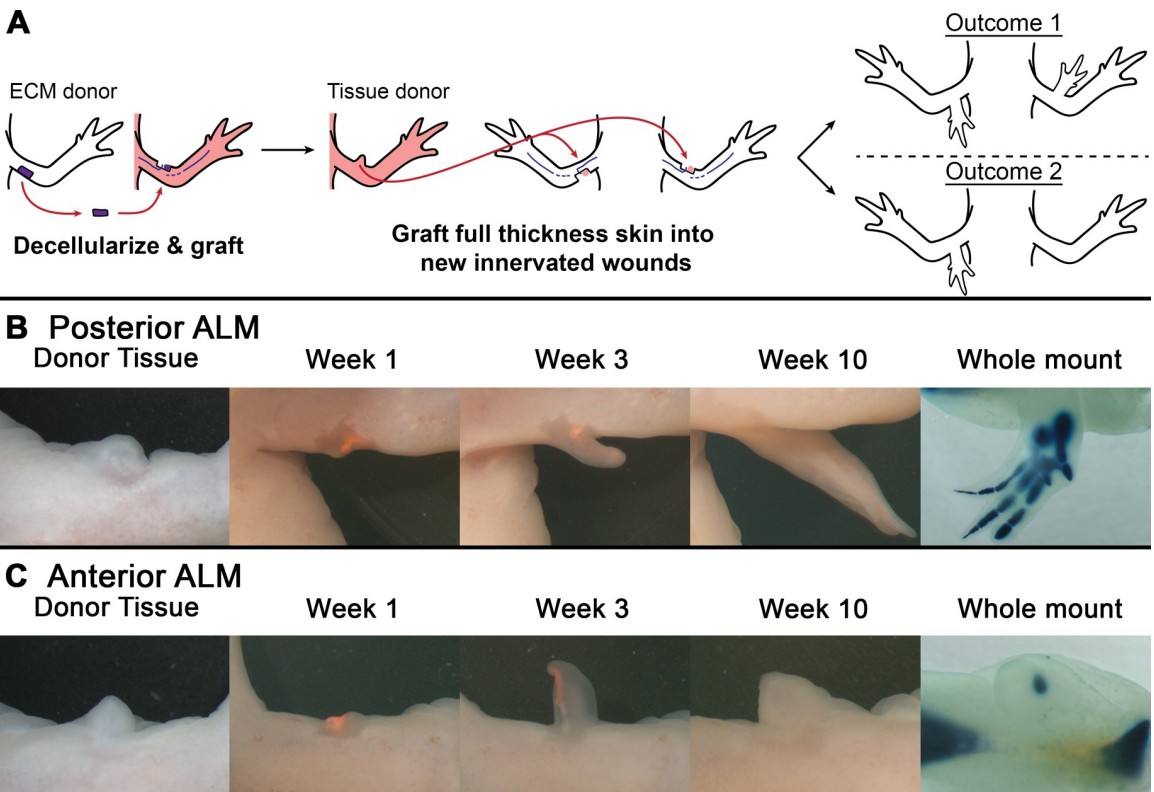

**Fig 2. ECM induced ectopic structures have anterior but not posterior positional information.** A) Experimental design. Initially, decellularized posterior ECM (purple) was implanted into anterior, innervated wound sites of transgenic fluorescent (GFP+ and RPF+) donor animals. The resultant ectopic structures were allowed to pattern and differentiate. Then, full thickness skin was harvested from the ECM-induced ectopic structures (donor tissue) and grafted into either anterior or posterior innervated wound sites of white host animals. If decellularized ECM grafts were sufficient to elicit the generation of new (non-anterior) positional identities in the tissue donor wounds, then grafts of this tissue into a new host will elicit a regenerative response in both anterior and posterior innervated wound sites (outcome 1). If the ECM grafts were insufficient to induce the formation of new positional identities in the tissue donor wounds, then grafts of this tissue will only elicit a regenerative response when implanted into posterior innervated wound sites (outcome 2). B-C) Representative time course of live images showing the most complex pattern elicited when full thickness skin from an ECM-induced ectopic structure (donor tissue) was implanted into an anterior or posterior located ALM. At 10 weeks, tissue was harvested and underwent whole mount cartilage staining to assess for skeletal patterning (last image in panel). B) Grafts into posterior located ALMs resulted in the generation of more complex pattern (9 out of 12) relative to anterior sites. C) Grafts into anterior located ALMs usually resulted in no ectopic growth (25 out of 29), but sometimes resulted in the generation of a single cartilage nodule (4 out of 29). For statistics relating to patterning phenotypes see Table 1. Red signal in all live images represents grafted RFP tissue.

Mid-bud stage blastema tissue, generated from anterior and posterior ALM sites in the absence of a tissue graft, were treated with either DMSO (vehicle control) or 150µg per gram body weight of RA (Fig 3A). Tissues were collected 7 days post treatment, as this time point has been shown to be sufficient to document significant alterations in the relative expression

**Table 1. Quantification of ectopic limb structures induced in anterior and posterior ALMs.**

| ALM location | Total ALMs performed | Successful ALMs surgeries* | No element | Single nodule | Multiple asymmetrical elements | Complete limb |
|---|---|---|---|---|---|---|
| Anterior | 34 | 29 | 25 (86.2%) | 4 (13.8%) | 0 | 0 |
| Posterior | 34 | 12 | 3 (25.0%) | 0 | 5 (41.7%) | 4 (33.3%) |

*only ALMs that generated a blastema and retained a tissue graft at 3 weeks post-surgery were used for subsequent analysis. Due to technical difficulties, posterior ALMs are prone to a higher rate of engraftment failure than anterior counterparts.

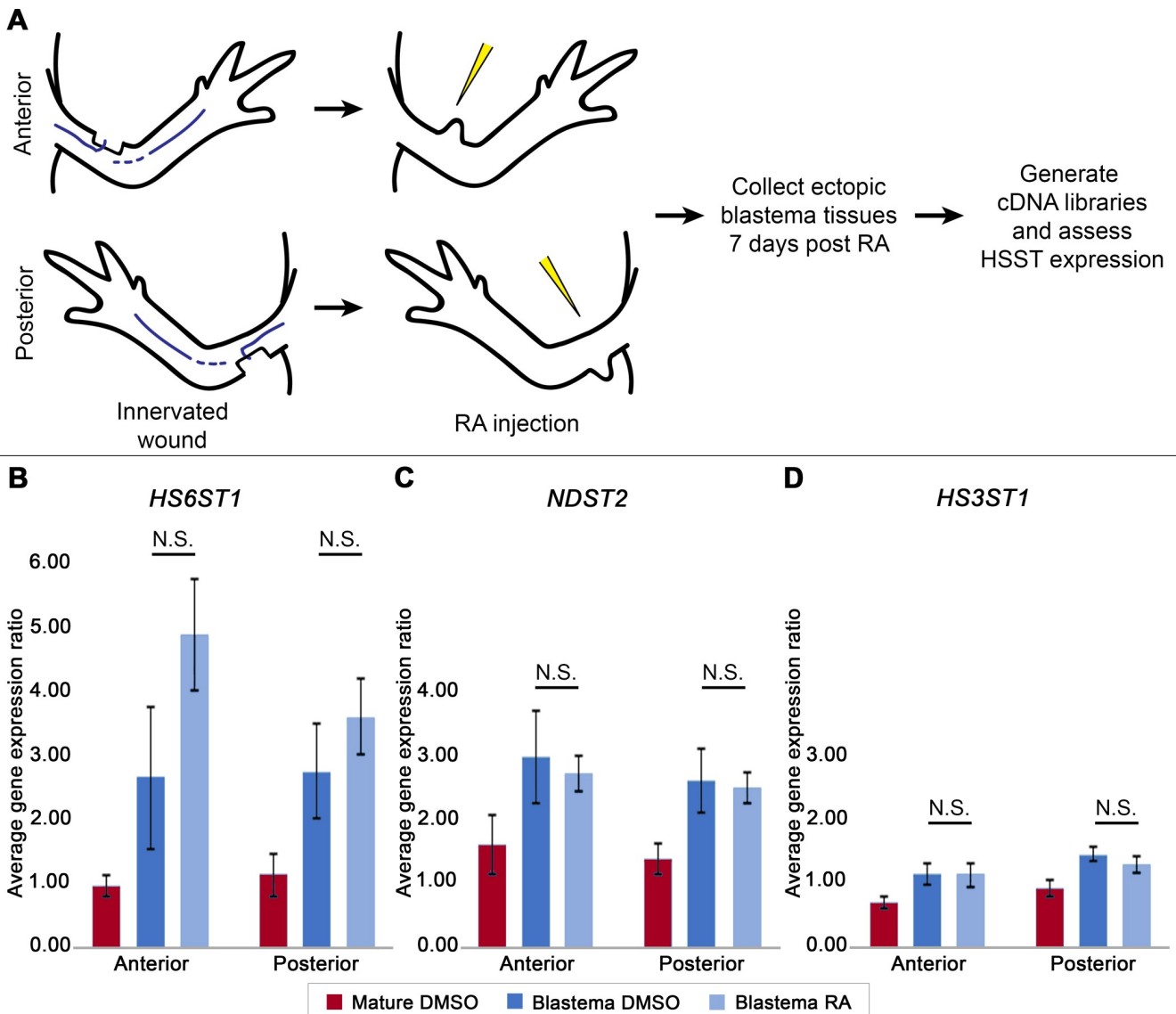

**Fig 3. Location specific expressional analysis of heparan sulfate sulfotransferases (HSSTs) in response to RA treatment in blastema and mature axolotl limb tissue.** A) Experimental design. Innervated wound sites were generated on the anterior and posterior limb axis of the axolotl forelimb. Once mid-bud stage blastema tissue had been generated at these sites, animals were treated with either DMSO (vehicle control) or RA. Blastema and mature tissues were collected 7 days post treatment and used for the generation of cDNA libraries and subsequent HSST transcriptional analysis. (B-D) Histograms representing *HS6ST1* (N = 4), *NDST2* (N = 3) and *HS3ST1* (N = 4) expression, relative to Ef-1α. Quantification of relative HSST gene expression was conducted using the Pfaffl method [36]. Error bars represent standard error of the mean. N.S. represents no statistical significance (p > 0.05; t-test).

of the patterning genes *Shh* and *Alx4* in anterior and posterior blastema tissue [7]. Transcriptional analysis by q-RTPCR was then performed for the 3 different HSST genes—*HS6ST1*, *NDST2* and *HS3ST1*. Our data revealed no significant difference in the relative expression of each of these HSST genes following RA treatment, irrespective of the limb location (Fig 3).

One surprising outcome to this expression analysis was that in the context of the ALM *HS6ST1*, *NDST2*, and *HS3ST1* are not differentially expressed in anterior and posterior ALMs. Previously described anterior/posterior expressional differences in HSSTs were observed in amputation blastemas [13], which are composed of connective-tissue derived blastema cells from the different limb axes. In contrast, the current expression study was performed on ALM

blastemas, which are generated by relatively "positionally pure" populations of connective tissue derived cells (i.e. only cells from anterior or posterior). Thus, it is possible that position specific differences in HSST expression (and HSPG composition) are downstream of pattern formation, and thus would require positionally diverse blastema cell populations.

Another important consideration is that in the current study we quantified gene transcription, and not protein level or function. It is plausible that translation or HSST or HSPG protein function is different across the limb axes, and RA treatment may somehow affect these processes. Future studies would be required to evaluation this. Never-the-less, we concluded that *HS6ST1*, *NDST2*, and *HS3ST1* transcriptional expression is not altered in blastema tissue in response to RA-directed positional reprogramming. We interpret this data to implicate that HSPGs are unlikely to play a direct role in communicating positional information during pattern formation, and thus would not be considered a Type 1 or Type 2 cue. However, as we will discuss below, it is probable that HSPGs play a supportive role in the transmission of Type 2 communications.

## 3. Discussion

### 3.1. The patterning hierarchy

Amphibian limb regeneration is orchestrated by a series of interconnected and temporally distinct signaling pathways, which activate specific molecular and biochemical processes. A thorough understanding of these processes and the necessary inputs is therefore required before attempting to successfully engineer an equivalent regenerative response in human tissue. Of particular interest in mounting an effective regenerative response is pattern formation, which specifies not only the cell types needed but also where the cells should be positioned in the three-dimensional structure. We propose that pattern formation during regeneration occurs in a hierarchical fashion [21].

At the first level of the hierarchy, pattern-forming cells have discrete and stabilized positional information. Experimental research demonstrates that cells of connective tissue origins retain this positional memory [4, 5, 12, 22, 23, 37, 38]. In response to amputation, a regenerative permissive environment is established and the migration of cells into this environment juxtaposes cells from the different limb axes at the site of injury. This facilitates the second level of the patterning hierarchy, where connective tissue derived cells from the different limb axes communicate their positional information (Type 1), which drives the formation of new cells with the missing positional information [16, 17]. The third and fourth levels of the hierarchy involves pattern forming cells establishing location specific molecular signaling centers that generate Type 2 positional cues which directs both pattern-forming (connective tissue) and–following (other cell types) cells as they organize into the tissues of the regenerate. These last two positions on the hierarchy do not involve the generation of any new positional information but instead are concerned with the interpretation and output of the Type 2 positional cues.

The patterning hierarchy model adequately explains the phenomenological differences documented between RA, SHH and FGF8 induced ectopic limbs in the ALM. RA reprograms positional information in receptive tissues (blastema tissue) to a posterior-ventral-proximal limb identity [7, 11, 25–29], and this change is stabilized in the cells. Therefore, RA affects the first level of the patterning hierarchy by changing the intrinsic positional information of the connective tissue cells and thus can elicit the generation of pattern from innervated wound sites (modified ALM) by facilitating the formation of new positional identities [11]. In contrast, SHH and FGF8 signaling both clearly instruct the organization of the regenerating limb

cells, but do not change the actual positional identity of these cells [12], and therefore function at the lower level of the patterning hierarchy as a Type 2 cue.

HSPGs modulates specific signaling pathways [13, 30, 34], including Hedgehog and FGF signaling, which directly contribute towards patterning output in developing and regenerating limbs [12, 13, 34]. However, our findings show that ECM-induced ectopic structures, which exhibit minimal pattern, lack positional identities outside of the original wound site. Since the grafted ECM tissue would include associated HSPGs, this suggests that none of these molecules constitutes a Type 1 positional communication. Moreover, we observed that HSST expression is unaffected by positional reprogramming to a proximal-posterior-ventral identity by RA treatment, while Type 2 molecules are affected [7]. Therefore, HSPGs are not directly responsible for the communication of positional information between cells with positional memory (Type 1), nor behave in a manner similar to molecules that mediate Type 2 positional communications. HSPGs may participate at the lower levels of the hierarchy (facilitating patterning output) by interacting with autocrine/paracrine signals that communicate pattern information, such as SHH and FGF8.

## 3.2. Communication of pattern inducing positional cues

There is a growing list of molecules which mediate Type 2 positional communications, including FGF8, SHH, and WNT5a. SHH and FGF8 elicit patterned formation but fail to establish new positional identities in the axolotl during regeneration [12]; and in human cell culture HOXA13-dependent WNT5a production by human foot derived fibroblasts facilitates palmo-plantar specialization of positionally naïve epithelium [22].

To date, the molecules that mediate Type 1 positional communications in the regenerate have yet to be established. The current study indicates that these molecules are not present in the ECM. One plausible hypothesis is that Type 1 communications occur through cell-cell interactions, facilitated by cell surface molecules. These interactions have already been suggested to contribute towards regenerative output [25, 39], and may include cell surface molecules such as cadherins (N-cadherin, desmoglein 4 preprotein) and integrins ($\alpha1$, $\alpha3$, $\alpha6$, $\alpha v$, $\beta1$, and $\beta3$), which exhibit differential expression within regenerating tissue, both spatially and temporally [40–44]. Although none of these molecules have been directly linked to positional information, it is speculated that cell adhesion molecules mediate the selective adhesion properties previously characterized in the blastema [25, 39].

Selective adhesion has been documented along the Proximal/Distal (P/D) axis of the amphibian limb. Late bud stage blastema tissue ends up at the appropriate location along the P/D limb axis if autographed to a more proximal regenerating limb site [25]. In *Notophtalmus viridescens*, Prod-1 is a membrane bound CD59-like molecule which exhibits a graded expression along the P/D axis of forelimb and has been implicated in this P/D selective adhesion phenomenon [39]. In the axolotl, blastema cells with altered Prod-1 expression relocate to new P/D locations [45]. However, unlike newt, axolotl Prod-1 is shed from the cell membrane [46]; suggesting that either a variety of molecules are involved in this selective adhesion process or that species specific differences are at play. It is presently unresolved whether selective adhesion is a means to communicate positional information, or if it is a result of differences in tissue identities along the P/D axis. Moreover, the identification of cell-surface molecules may mediate Type 1 communications along the Anterior/Posterior limb axis is lacking.

Due to this lack of understanding in the communication of positional identity, future research in the field should assess for discrete properties and/or cell-surface molecules associated with pattern forming (connective tissue) cells from different anatomical positions within the amphibian limb in order to identify potential candidates mediating Type 1 positional

communications during regeneration-associated pattern formation. Once such communications systems have been identified in the axolotl, the presence of these systems in other regenerating and non-regenerating systems can be sought to assess for conservation during evolution.

## 4. Experimental procedures

### 4.1. Animal husbandry

Surgical manipulations and tissue collections were performed on white-strain (RRID: AGSC_101J), GFP strain (RRID: AGSC_110J), and RFP strain (RRID: AGSC_112J) Mexican axolotls, spawned on site at UMass Boston or obtained from the Ambystoma Genetic Stock Center at the University of Kentucky. Anesthetization prior to surgery was administered by a 0.1% MS222 solution (Ethyl 3-aminobenzoate methanesulfonate salt, Sigma), pH 7.0.

All experimental work was approved by the Institutional Animal Care and Use Committee of the University of Massachusetts, Boston (protocol number: IACUC 2015004) and conducted in accordance with the recommendations in the Guide for the Care and Use of Laboratory Animals of the National Institutes of Health.

### 4.2. Surgical methods

**4.2.1. ECM preparation.** Posterior, decellularized ECM was prepared according to Phan et al., [13]. Full thickness posterior, stylopod skin was collected from white animals and rinsed in 60% Dulbecco's Modified Eagle's Medium (DMEM, sigma). After brief sterilization in 70% ethanol, the tissue was incubated under gentle agitation three times for 3 minutes in $Ca^{2+}/Mg^{2+}$ free Hanks solution and the epidermis was peeled off with forceps. The tissue was incubated in 2M urea solution twice for 15 minutes under gentle agitation and then rinsed and stored in 80% PBS for 2 days before grafting into innervated wound sites.

**4.2.2. ECM induction of ectopic structures.** Phan et al., [13] first described this method, a modified version of the ALM [13]. In the current study, 2mm x 2mm lateral wounds were generated on the anterior side of the stylopod of GFP and RFP animals (approximately 14cm snout to tail tip). The brachial nerve was deviated to the wound site, and the animals were kept on ice post-deviation for 1 and half hours to promote nerve retention in the site. The innervated wounds were allowed to heal for 2 days, facilitating wound epithelium formation, and then decellularized posterior ECM (see method 5.2.1) was grafted into the wound site, secured under the newly formed wound epithelium. Post-surgery, the animals were maintained on ice for 2 hours to enhance healing and retention of the ECM graft. The wound sites were monitored closely for the next 3 weeks, and any site that lost the grafted tissue or failed to generate a blastema in this time was excluded from the final analysis.

**4.2.3. Amputation of and tissue collection from ECM-induced ectopic structures.** Eight weeks post ECM implantation, the ectopic structures were considered fully patterned as they were covered by fully developed skin. Approximately 50% of each of these structures were amputated and monitored, on a weekly basis for 8 weeks, for regenerative output. Full thickness skin was collected from the distal end of the ECM-induced ectopic structures and implantation into anterior and posterior innervated wounds (see section 5.2.4).

**4.2.4. The Accessory Limb Model (ALM).** To determine if posterior ECM was sufficient to generate new positional identities in anterior wound sites, tissue was collected from the distal end of ECM-induced structures on GFP and RFP animals and implanted into anterior and posterior ALMs. The ALM was originally described by Endo et al. [4]. Briefly, a lateral wound is generated on the axolotl forelimb stylopod by removing a 2mm x 2mm piece of full thickness skin. In the current study, this was performed on either the anterior or posterior of the limb.

Innervation was achieved by deviating the brachial nerve to the wound site and the animals were maintained on ice for 1 and half hours post-surgery to promote nerve retention. Two days later, once a wound epithelium had been generated, the collected tissue (see section 5.2.3) was implanted into the wound site, under the wound epithelium. Post-grafting, the animals were maintained on ice for 1 hour and 30 minutes to promote graft retention. If a positional disparity was established between the wound site and the tissue graft, pattern formation would be facilitated and lead to regenerative output. Absence of a positional disparity would lead to blastema regression.

The wound sites were monitored on a weekly basis for a regenerative response. Over the first 3 weeks, any site that lost the tissue graft or failed to generate a blastema was excluded from the final analysis. Fluorescent assessment in the live animals allowed for tracking of the grafted tissue in the wound sites. Regeneration was considered to be complete (patterned and differentiated) when the ectopic structures was covered by fully developed skin.

**4.2.5. Reprogram of HSST gene expression.** To determine if RA was sufficient to posteriorize HSST gene expression in blastema tissue, a modified version of the ALM was used [7, 11]. Here, innervated anterior and posterior wounds were generated on white animals (approximately 5–8 cm snout to tail tip), in the absence of a tissue graft. Once mid-bud stage blastemas had formed (9 days post-surgery) the animals were treated with RA or vehicle (DMSO). RA was administered intraperitoneally at 150μg per gram of body weight [11, 47], and animals were kept in the dark for 2 days after injection to minimize photoinactivation of the drug. 7 days post treatment, blastema and mature limb (full thickness skin) tissues were collected, extracted, and stored in Tripure Isolation Reagent (Sigma). 4 blastema or mature tissue samples were pooled and used to generate a single biological replicate. 4 biological replicates were produced per treatment and tissue type.

## 4.3. Whole mount staining and phenotype scoring

Whole mount cartilage staining with Victoria blue was conducted as described in Bryant and Iten et al., 1974 [48]. The ectopic cartilage formed in the ALM wound sites were scored based upon the number of skeletal elements documented. The following classifications were used, as described in [7]—no element, single nodule, multiple symmetrical elements, multiple asymmetrical elements and complete Limb.

## 4.4. mRNA isolation, cDNA synthesis, and differential gene analysis by qPCR

mRNA was isolated from tissue samples extracted in Tripure Isolation Reagent using the Nucleospin RNA XS kit (Macherey-Nagel) according to the manufacturers' specifications. The

**Table 2. Primer sequences utilized for differential analysis of HSSTs in axolotl limb tissue.**

| Name | Sequence |
|---|---|
| Ef-1α Forward | 5′–AACATCGTGGTCATCGGCCAT |
| Ef-1α Reverse | 5′–GGAGGTGCCAGTGATCATGTT |
| HS6ST1 Forward | 5′—ACGCCTGACCCACACTACGTCA |
| HS6ST1 Reverse | 5′—CCAGGCGCACATTTTGTACCAGGT |
| NDST2 Forward | 5′—CGTCTTTGCCTGACACTTGA |
| NDST2 Reverse | 5′—ACGTGAAGTTGGGAACCAAG |
| HS3ST1 Forward | 5′—TATTAACATGTCGCCGTCCA |
| HS3ST1 Reverse | 5′—TCCGTCGAAAAACTTCTGCT |

mRNA was then converted into cDNA libraries using the Transcription first strand cDNA synthesis kit (Roche), in conjunction with the anchored oligo(dT)18 primer, as specified by the manufacturer. Transcription of HSST genes, relative to that of the housekeeping gene *Ef-1α*, in the different experimental samples was assessed by qPCR (Azuraquant™ Green Fast qPCR Mix Lo-Rox, Azura Genomics Inc.), using the Pfaffl method [36]. Primer sequences used are presented in Table 2, including the *HS6ST1* primer set which was previously published by Phan et al. [13].

## Supporting information

**S1 Fig. All induced and amputated ECM-induced ectopic structures.** Ectopic structures were induced by grafting decellularized ECM into anterior, innervated wound sites on axolotl forelimbs. Of the 50 anterior ALM sites implanted with posterior derived ECM, live-assessment showed 37 with small, rounded or digit like outgrowths by week 9. At week 13, 34 of the ectopic growths were large enough to amputate and these structures were allowed to regenerate. Amputated structures were monitored for 8 weeks for a regenerative response. 3 of the 34 amputated structures exhibited the formation of a blastema but resulted in no regenerated pattern. The remaining 31 amputated structures simply healed over. Once regeneration/healing was completed, limbs were collected and underwent whole mount cartilage staining.
(TIF)

**S2 Fig. Full thickness skin derived from ECM-induced ectopic structures elicit complex pattern when grafted into posterior located ALMs.** Grafts, derived from ECM-induced ectopic structures, into posterior located ALMs resulted in the generation of complex pattern (9 out of 12). For statistics relating to patterning phenotypes see Table 1. Red and green signal in all live images represents grafted RFP and GFP tissue respectively. Only ALMs that generated a blastema and retained a tissue graft at 3 weeks post-surgery were used for analysis in this study. Once regeneration was completed, limbs were collected and underwent whole mount cartilage staining.
(TIF)

**S3 Fig. Full thickness skin derived from ECM-induced ectopic structures elicit very little pattern when grafted into anterior located ALMs.** Grafts, derived from ECM-induced ectopic structures, into anterior located ALMs frequently resulted in no ectopic growth (25 out of 29), but sometimes resulted in the generation of a single cartilage nodule (4 out of 29). For statistics relating to patterning phenotypes see Table 1. Red and green signal in all live images represents grafted RFP and GFP tissue respectively. Only ALMs that generated a blastema and retained a tissue graft at 3 weeks post-surgery were used for analysis in this study. Once regeneration was completed, limbs were collected and underwent whole mount cartilage staining.
(TIF)

**S1 Data. Excel spreadsheet with all the raw and processed data for the evaluation of the transcriptional expression of *HS6ST1*, *NDST2* and *HS3ST1* in location specific blastema tissue, in the presence and absence of RA.**
(XLSX)

## Acknowledgments

The authors would like to acknowledge Dr. Anne Phan for her insightful discussions regarding this work.

## Author Contributions

**Conceptualization:** Warren A. Vieira, Catherine D. McCusker.

**Data curation:** Warren A. Vieira, Shira Goren.

**Formal analysis:** Warren A. Vieira, Shira Goren.

**Funding acquisition:** Catherine D. McCusker.

**Investigation:** Warren A. Vieira, Shira Goren, Catherine D. McCusker.

**Methodology:** Warren A. Vieira, Catherine D. McCusker.

**Project administration:** Catherine D. McCusker.

**Supervision:** Warren A. Vieira, Catherine D. McCusker.

**Visualization:** Warren A. Vieira.

**Writing – original draft:** Warren A. Vieira.

**Writing – review & editing:** Warren A. Vieira, Catherine D. McCusker.

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
