## [Decision Letter · Decision Letter 0]

2 Feb 2021

PONE-D-20-40259

ECM-mediated positional cues are able to induce pattern, but not new positional information, during axolotl limb regeneration

PLOS ONE

Dear Dr. McCusker,

Thank you for submitting your manuscript to PLOS ONE. After careful consideration, we feel that it has merit but does not fully meet PLOS ONE’s publication criteria as it currently stands. Therefore, we invite you to submit a revised version of the manuscript that comprehensively addresses the points raised during the review process.

We look forward to receiving your revised manuscript.

Kind regards,

Michael Schubert

Academic Editor

PLOS ONE

Reviewers' comments:

Reviewer's Responses to Questions

**Comments to the Author**

1. Is the manuscript technically sound, and do the data support the conclusions?

Reviewer #1: Partly

Reviewer #2: Yes

2. Has the statistical analysis been performed appropriately and rigorously? 

Reviewer #1: Yes

Reviewer #2: Yes

3. Have the authors made all data underlying the findings in their manuscript fully available?

Reviewer #1: Yes

Reviewer #2: Yes

4. Is the manuscript presented in an intelligible fashion and written in standard English?

Reviewer #1: Yes

Reviewer #2: Yes

5. Review Comments to the Author

Reviewer #1: The authors expand upon prior work (Pham et al 2015) and a hypothesis (Vieira and McCusker 2019) to test the hypothesis that extracellular matrix materials, specifically the heparan sulfate proteoglycans (HSPGs), play a role in the upper tier of decisions that pattern limbs. They reject the hypothesis. The data and interpretation is thought-provoking and worthy of publication, but requires a little more careful exposition and qualification.

Of the three component parts of the study, I query the first. The Accessory Limb Model (ALM) was an interesting choice to test the hypothesis using matrix denuded of cells as graft material. Herein lies a flaw - there is no assurance that matrix retains the incremental compositional or constitutional differences once transplanted. Unlike transplants with both matrix and cells, matrix alone does not elicit anything other than a very rudimentary hypomorphic structure. Indeed, it would be hard to state with conviction that a single cartilaginous nodule represents epimorphic versus tissue regeneration or even some sort of unusual scar cicatrix. To be definitive, the study would have needed to show that matrix retains its HSPG characteristics well beyond would healing and into blastema patterning stages. That said, I think the data is indicative, even highly suggestive, but not definitive. The loss of posterior coding in hypomorph blastema used for ALM grafts may merely reflect the gradual loss of posterior ECM features post-implantation: weeks (8 weeks of hypomorphic - ie; unusual - growth) are eons in inflamed tissues that are undergoing radical matrix (re)modelling and proliferation. I am not convinced that the mature ALM graft tissue retains or even reflects testable posterior origins that much later. You'd need mass spec, or better still some sort of immunohistochemistry to be sure that posterior attributes remain static. That said, I think these experiments have a place in the paper, but interpretation needs to be more carefully discussed and qualified.

The manuscript would benefit from some editing to shorten sentences, economize on verbiage, and to correct minor grammatical errors: it was strangely difficult to understand the flow of experimental design regarding the ALM portion, and I am not sure why. Perhaps this is a feature of my COVID isolation-induced hazy state of mind...? Perhaps a flow chart encompassing all facets of the approach would help clarify? (ant/posterior source amputation result, transplant source/engraftment site process and result - the diagram in fig 2 helped, but didn't quite get me there)?

Trivial small editorial stuff:

p 6, - 2nd sentence from bottom - the semi-colon should be a comma

p 15 under 4.2 - "Downstream cues function... ... and is communicated" should read ARE communicated (since the subject of the sentence is "cues"

p 17 under 5.2.1, third line down "gentle agitate" should read gentle agitation.

Reviewer #2: Author´s utilized the ALM-protocol, to generate ECM-induced structures and to show that HSPGs play a role as a molecular signal which downstream communicates information for the proper patterning during axolotl limb regeneration.

Experiments are technically sound and the conclusions are coherent with the findings. I havo no major critics on the content of the manuscript. However I believe that putting more effort in the graphic work presented in the main figures, as well as the design and presentation of the figures will improve the manuscript.

Besides this point, I think the manuscript is well written and has the merits to be published in PlosOne.

6. PLOS authors have the option to publish the peer review history of their article (what does this mean?). If published, this will include your full peer review and any attached files.

Reviewer #1: No

Reviewer #2: No

---

## [Author Response · Author response to Decision Letter 0]

12 Feb 2021

Response to Reviewers

We would like to begin by thanking the editor and reviewers for their time and input into the review of our manuscript. The valuable feedback provided has helped to strengthen the quality of document. We have worked through the comments provided by the reviewers and made the appropriate changes to the text and figures. Below we will address each of the comments specifically (our responses provided as bolded text), detailing how the manuscript was modified to resolve the raised concerns. 

Reviewer #1 Comments

The authors expand upon prior work (Pham et al 2015) and a hypothesis (Vieira and McCusker 2019) to test the hypothesis that extracellular matrix materials, specifically the heparan sulfate proteoglycans (HSPGs), play a role in the upper tier of decisions that pattern limbs. They reject the hypothesis. The data and interpretation is thought-provoking and worthy of publication, but requires a little more careful exposition and qualification.

Of the three component parts of the study, I query the first. The Accessory Limb Model (ALM) was an interesting choice to test the hypothesis using matrix denuded of cells as graft material. Herein lies a flaw - there is no assurance that matrix retains the incremental compositional or constitutional differences once transplanted. Unlike transplants with both matrix and cells, matrix alone does not elicit anything other than a very rudimentary hypomorphic structure. Indeed, it would be hard to state with conviction that a single cartilaginous nodule represents epimorphic versus tissue regeneration or even some sort of unusual scar cicatrix. To be definitive, the study would have needed to show that matrix retains its HSPG characteristics well beyond would healing and into blastema patterning stages. That said, I think the data is indicative, even highly suggestive, but not definitive. The loss of posterior coding in hypomorph blastema used for ALM grafts may merely reflect the gradual loss of posterior ECM features post-implantation: weeks (8 weeks of hypomorphic - ie; unusual - growth) are eons in inflamed tissues that are undergoing radical matrix (re)modelling and proliferation. I am not convinced that the mature ALM graft tissue retains or even reflects testable posterior origins that much later. You'd need mass spec, or better still some sort of immunohistochemistry to be sure that posterior attributes remain static. That said, I think these experiments have a place in the paper, but interpretation needs to be more carefully discussed and qualified.

Response: We thank the reviewer for this feedback. These comments helped us pinpoint places in the text that needed clarification, which we have worked to correct in the revised manuscript. The premise of our study was to determine if ECM was sufficient, on its own, to elicit the generation of new positional identities in a regeneration competent environment. 

Our experimental design (using amputation, grafting and transcription-based assays) was to test whether posterior ECM, when grafted into anterior innervated wound sites, was sufficient to elicit a cellular growth response that could generate new, non-anterior, positional identities and, in turn, the formation of ectopic limb structures. 

The aim of our experiments was not to determine whether posterior ECM could retain posterior positional information for prolonged periods, when grafted into new anatomical locations. Rather, we were testing its ability to elicit the formation of new positional information from the host cells. Based on the patterning hierarchy model, new positional information is generated early on in the regenerative process and, therefore, ECM-associated positional information cues, if present, would not require long term stability in this context. We hope that out modifications to the text and figures will help to clarify this to the reader. 

The manuscript would benefit from some editing to shorten sentences, economize on verbiage, and to correct minor grammatical errors: it was strangely difficult to understand the flow of experimental design regarding the ALM portion, and I am not sure why. Perhaps this is a feature of my COVID isolation-induced hazy state of mind...? Perhaps a flow chart encompassing all facets of the approach would help clarify? (ant/posterior source amputation result, transplant source/engraftment site process and result - the diagram in fig 2 helped, but didn't quite get me there)?

Response: We thank Reviewer #1 for this feedback. We apologize for the difficulty in the understanding associated with our experimental design. To remedy these raised concerns, we first worked through the manuscript to economize the text. Next, we correct any grammatical errors present. All changes have been tracked and presented in the ‘Revised Manuscript with Track Changes’ document. Lastly, to improve the understanding of our experimental design, we redesigned figures 1 and 2 (both associated with ALM experiments). 

For figure 1, we included an experimental outline (new panel A). This describes each step of the experiment. This panel also includes two potential outcomes for the experiment - either the structures regenerate or not. We also explain the implications of each outcome in the legend of the figure. If the structures regenerate, then ECM was sufficient to generate new, non-anterior positional information in the innervated would sites. If the structures do not regenerate, then ECM is insufficient to elicit the formation of new positional information. 

For figure 2, we redesigned panel A. The experimental outline now includes two potential outcomes for the experiment - either regenerative responses are elicited in both anterior and posterior host sites, or only in the posterior sites. We also explain the implications of each outcome in the legend of the figure. If ectopic pattern is generated in both anterior and posterior host sites, then the donor tissue comprised of anterior and non-anterior positional identities. Therefore, the grafted ECM was sufficient to generate new, non-anterior positional information in the original innervated would sites. If ectopic pattern is only generated in the posterior host sites, then the donor tissue comprised only of anterior identities. In this situation, ECM was insufficient to elicit the generation of new positional information in the original wound sites. 

Trivial small editorial stuff:

p 6, - 2nd sentence from bottom - the semi-colon should be a comma

p 15 under 4.2 - "Downstream cues function... ... and is communicated" should read ARE communicated (since the subject of the sentence is "cues"

p 17 under 5.2.1, third line down "gentle agitate" should read gentle agitation.

Response: These have all been corrected.

Reviewer #2 Comments

Author´s utilized the ALM-protocol, to generate ECM-induced structures and to show that HSPGs play a role as a molecular signal which downstream communicates information for the proper patterning during axolotl limb regeneration.

Experiments are technically sound and the conclusions are coherent with the findings. I havo no major critics on the content of the manuscript. However I believe that putting more effort in the graphic work presented in the main figures, as well as the design and presentation of the figures will improve the manuscript.

Besides this point, I think the manuscript is well written and has the merits to be published in PlosOne.

Response: We thank Reviewer #2 for this feedback. We have made several changes to the figures to help improve the understanding associated with them and our study. The redesigned figures, and corresponding changes to the legends, have been highlighted in the revised manuscript.

---

## [Decision Letter · Decision Letter 1]

19 Feb 2021

ECM-mediated positional cues are able to induce pattern, but not new positional information, during axolotl limb regeneration

PONE-D-20-40259R1

Dear Dr. McCusker,

We’re pleased to inform you that your manuscript has been judged scientifically suitable for publication and will be formally accepted for publication once it meets all outstanding technical requirements.

Kind regards,

Michael Schubert

Academic Editor

PLOS ONE

Reviewer's Responses to Questions

**Comments to the Author**

1. If the authors have adequately addressed your comments raised in a previous round of review and you feel that this manuscript is now acceptable for publication, you may indicate that here to bypass the “Comments to the Author” section, enter your conflict of interest statement in the “Confidential to Editor” section, and submit your "Accept" recommendation.

Reviewer #2: All comments have been addressed

2. Is the manuscript technically sound, and do the data support the conclusions?

Reviewer #2: Yes

3. Has the statistical analysis been performed appropriately and rigorously? 

Reviewer #2: Yes

4. Have the authors made all data underlying the findings in their manuscript fully available?

Reviewer #2: Yes

5. Is the manuscript presented in an intelligible fashion and written in standard English?

Reviewer #2: Yes

6. Review Comments to the Author

Reviewer #2: I am satisfied with the new version of the manuscript, the new version of the figures. In my opinion this manuscript is ready for publication in Plos One.

7. PLOS authors have the option to publish the peer review history of their article (what does this mean?). If published, this will include your full peer review and any attached files.

Reviewer #2: No

---

## [Editor Report · Acceptance letter]

23 Feb 2021

PONE-D-20-40259R1 

ECM-mediated positional cues are able to induce pattern, but not new positional information, during axolotl limb regeneration 

Dear Dr. McCusker:

I'm pleased to inform you that your manuscript has been deemed suitable for publication in PLOS ONE. Congratulations! Your manuscript is now with our production department. 

Kind regards, 

on behalf of

Dr. Michael Schubert 

Academic Editor

PLOS ONE